# Utilization of Functionalized Metal–Organic Framework Nanoparticle as Targeted Drug Delivery System for Cancer Therapy

**DOI:** 10.3390/pharmaceutics15030931

**Published:** 2023-03-13

**Authors:** Vy Anh Tran, Van Thuan Le, Van Dat Doan, Giang N. L. Vo

**Affiliations:** 1Institute of Applied Technology and Sustainable Development, Nguyen Tat Thanh University, Ho Chi Minh City 700000, Vietnam; 2Faculty of Environmental and Food Engineering, Nguyen Tat Thanh University, Ho Chi Minh City 700000, Vietnam; 3Center for Advanced Chemistry, Institute of Research and Development, Duy Tan University, 03 Quang Trung, Da Nang 550000, Vietnam; 4Faculty of Natural Sciences, Duy Tan University, 03 Quang Trung, Da Nang 550000, Vietnam; 5The Faculty of Chemical Engineering, Industrial University of Ho Chi Minh City, Ho Chi Minh City 700000, Vietnam; 6Faculty of Pharmacy, University of Medicine and Pharmacy at Ho Chi Minh City, Ho Chi Minh City 700000, Vietnam

**Keywords:** metal–organic frameworks, targeting drug delivery, targeted cancer therapy, light response, multi-targeted response

## Abstract

Cancer is a multifaceted disease that results from the complex interaction between genetic and environmental factors. Cancer is a mortal disease with the biggest clinical, societal, and economic burden. Research on better methods of the detection, diagnosis, and treatment of cancer is crucial. Recent advancements in material science have led to the development of metal–organic frameworks, also known as MOFs. MOFs have recently been established as promising and adaptable delivery platforms and target vehicles for cancer therapy. These MOFs have been constructed in a fashion that offers them the capability of drug release that is stimuli-responsive. This feature has the potential to be exploited for cancer therapy that is externally led. This review presents an in-depth summary of the research that has been conducted to date in the field of MOF-based nanoplatforms for cancer therapeutics.

## 1. Introduction

To maximize therapeutic efficacy and minimize side effects, a substantial amount of work has been devoted to the development of novel micro- or nano-platforms for regulated and smart drug release systems, thanks in large part to the explosive expansion of materials chemistry [1]. High interest has been drawn to metal–organic frameworks (MOFs) ever since they were first reported in 1989 by Hoskins and Robson [2]. These materials are made by merging metal clusters or metal ions with organic ligands via coordinative bonds, resulting in a two- or three-dimensional topology that offers architectural control at the molecular scale. There are already over 20,000 unique MOF frameworks documented in the Cambridge database [3]. A wide range of metal–organic frameworks (MOFs) with tailored physicochemical properties (e.g., hydrophobicity or hydrophilicity, morphology, pore diameter, surface area) can be fabricated for critical applications, including separations [4], gas storage [5], analytical chemistry [6], catalysis [7,8,9], sensing [10], energy [11], imaging, and biomedicine [12].

A multifunctional nanomaterial is not merely an improved form of the initial capability [13]. In fact, multifunctional nanostructures incorporate multiple functions into a single particle to increase the carrier’s utility [14]. This layout has the potential to improve cancer treatment by allowing for a more precise detection, scanning, and management of the tumor’s microenvironment [15]. Multifunctional particles’ architectural backbones can be either organic (polymeric, liposomal, or proteinaceous) or inorganic (metal, nonmetallic, or biomimetic) [16,17,18]. The building blocks of peptide nanoparticles are amino acids, either organic or synthetic, that have been scaled down to the nanometer range. Due to their biodegradability, peptide nanoparticles have found widespread usage in cancer therapy, gene transfer, and target medication delivery [17]. However, while peptide-based delivery methods have shown promise in in vitro and in vivo research, this progress has been slower in transferring to clinical studies. Peptides are typically used for their therapeutic effects rather than as a delivery mechanism in peptide-based therapeutics [19]. In another advancement, biomaterials called metal–organic frameworks (MOFs) are created when metal cations or clusters are coordinated with organic ligands. MOFs have a large specific surface area and a high porosity, both of which promote greater contact with the cell membranes, leading to an increase in the amount of MOFs that are taken up by cells. Furthermore, positively charged MOF materials are able to connect to cell membranes via electrostatic interactions, and, as a result, are able to enter cells via the process of endocytosis [18]. MOFs delivery systems are on the verge of becoming one of the greatest promising approaches for use in biomedical applications.

The last decade has seen a massive increase in the utilization of MOFs in biological applications in terms of their fine-tunability, vast surface areas, and high loading capacities. In particular, a variety of applications for the delivery of drugs using MOFs are being investigated [12,20]. Initially, MOFs were employed to deliver medications in the form of tiny molecules, but more recent research has concentrated on the delivery of macromolecular cargo, including nucleic acids and proteins. Here, we explore the recent utilization of drug-delivery MOFs, with an emphasis on the alternatives that can be employed to build toward particular drug-delivery MOF applications. Among these choices are the MOF structure, the synthesis process, and the drug loading. Tuning, alterations, cellular targeting, biocompatibility, and uptake are other factors to take into account [12,21].

To battle diseases such as cancer, scientists are constantly looking for novel treatments, early diagnoses, and early detection methods. The principal application of MOFs beyond this viewpoint is their potential as cutting-edge materials and systems for cancer therapy [22]. Also highlighted are several difficult and promising facets of MOF-based cancer diagnosis and therapy. There are also a few successful case studies that can provide a phase change to clinics, but this is a fascinating field of science with progressive breakthroughs that require intense emphasis to fully transfer from bench to bedside. This is a crucial step in identifying the restrictions and barriers to the use of cutting-edge materials, such as MOFs, for the treatment of cancer that has reached the clinical stage [21,23].

Cancer is a leading cause of human mortality and poses a risk to nearly every family on Earth. In 2020, the number of people diagnosed with cancer was 19.3 million, and approximately 10 million lost their lives to the disease [24]. Cancer treatment typically consists of early diagnosis with X-ray computed tomography imaging, optical imaging, magnetic resonance imaging, and other imaging methods applicable to biological subjects, and late treatment with radiotherapy, chemotherapy, surgery, gene therapy, immunotherapy, and combination therapy. Chemotherapy, along with cytoreductive surgery and radiotherapy, is the most prevalent treatment for cancer. In particular, an abundance of immunotherapeutic and chemotherapeutic medications has been established and approved to be utilized by the Food and Drug Administration of the United States (FDA) because of the rising cancer rate. Nanotechnology devices are on par in size with macromolecules (such as enzymes and receptors) found in living organisms. Due to their microscopic size, nanoscale devices can easily interact with biomolecules and can also reach places previously inaccessible in the body, expanding the possibilities for illness detection and therapy [25]. Highly effective modulators of biomolecules have been produced by chemical biologists; however, many of these may not demonstrate perfect functionality in vivo because of an inadequate stability, solubility, poor pharmacokinetics, biocompatibility, and/or off-target activity. Scientists working in the field of nanotechnology have created devices to address the problems associated with the transport and pharmacokinetics of poorly behaving molecules. These devices work by enclosing active cargos and directing them into certain tissues, cells, or organelles [26].

Recently, MOFs have been developed as nanocarriers of medicines for cancer treatment [27,28]. The advantages of MOFs are their huge specific surface area, high porosity, and size controllability. They may be utilized as photothermal agents, photosensitizers, and Fenton reaction catalysts in chemodynamic therapy (CDT), photodynamic therapy (PDT), and photothermal therapy (PTT) (Figure 1). The poly(acrylic acid-mannose acrylamide) that further functionalized MOF-808 provided highly effective selective drug delivery with high cytotoxicity in HepG2 human hepatocellular carcinoma cells [14]. To increase the biocompatibility, extend blood circulation time, and target the encapsulated medication to the folate-expressing MCF-7 breast cancer, UiO-66 nanoparticles and folate-conjugated pluronic F127 were integrated [29]. Encapsulated in ZIF8 and loaded onto gelatin nanofibrous, phenamil, an activator of bone morphogenetic protein pathways, can kill MG-63 cells in vitro and suppress the formation of subcutaneous tumors in vivo [30]. In addition, MOFs can be loaded with therapeutic agents (drugs, photothermal agents, photosensitizers, etc.) for use in CT, PTT, PDT, CDT, and other tumor-specific treatments. In this article, we will discuss the most up-to-date findings about therapy-targeted MOFs as a foundation for cancer therapy.

This review presents an overview of various innovative nanomaterials generated for research and clinical application, highlights existing limitations and barriers that prevent the transfer from research to clinical usage, and addresses solutions for a more efficient utilization of nanoparticles in cancer therapy (Figure 2).

## 2. Basic Nanomaterial and Cancer and Target Therapy

### 2.1. Basics of Nanomaterials for Drug Delivery

Nanotechnology holds great promise for the management of chronic human diseases by delivering precise drugs to designated areas and targets. Recent years have seen many significant applications for the use of nanomedicine (biological, chemotherapeutic, and cancer immunotherapy agents) in the medical care of many diseases (Table 1). The review article offers a comprehensive summary of recent developments in the field of nanocarriers and drug delivery via nanocarrier technologies through a full assessment of the exploration and usage of nanostructured materials in enhancing both the effectiveness of old and new drugs and specific diagnosis through disease marker molecules [31].

Treatment delivery using MOFs has also been researched, starting with loading cancer drugs and controlling release. Small molecule medications, including the anticancer agents’ doxorubicin and curcumin, are still the main targets of applications of drug delivery MOFs. The development of macromolecular drugs with various MOFs, including plasmids, gelonin, siRNAs, and sgRNA-loaded Cas9 for CRISPR, has been accomplished via a method known as biomimetic mineralization. A burgeoning field of research focuses on using MOFs to deliver therapies of all sorts, such as cells, proteins, small chemicals, nucleic acids, gasotransmitters, and viruses [41].

### 2.2. Basics of Cancer and Target Therapy

The evolutionary lens is reshaping our knowledge of cancer. Peter Nowell was an early pioneer of the theory of tumor evolution. According to Nowell’s concept, most cancers begin with a single neoplastic cell and then progress through a process of selection for somatic modifications, with the most aggressive clones eventually proliferating and surviving [42]. Incredible advances in genetics and cell biology in recent years are shedding new light on cancer that contradicts prior conceptions. The evolutionary viewpoint provides five fundamentals of evolution necessary for understanding cancer: (1) the formation of malignancies takes place through a process called somatic selection; (2) ecological principles can be used to describe the relationship between tumors and microenvironments; (3) principles of behavioral ecology give insight on the dynamics in which cancer clones interact with one another in terms of cooperation and competition; (4) natural selection may be credited for the rarity of cancer; and (5) the common occurrence of cancer is explained by evolutionary medicine [43,44,45,46,47]. Cancer is therefore a multifaceted disease due to its complex combination of genetic and environmental factors. It is now known that DNA damage is a fundamental cause of the abnormalities that eventually lead to cancer. As a result of these alterations, uncontrolled cell division occurs, which ultimately leads to tissue damage (Figure 3).

To combat the development and spread of cancer, scientists have developed targeted cancer medicines that work by interfering with certain molecules (“molecular targets”). Various terms, including “molecule-targeted pharmaceuticals”, “molecule targeted therapies,” “precision medicines,” and others, are used to refer to targeted cancer treatments [48,49]. The high degree of specificity achieved by targeted therapy is remarkable. This specificity permits the following comparisons between targeted therapies and conventional chemotherapy [49]:In contrast to standard chemotherapies, which affect both rapidly dividing normal and malignant cells, targeted therapies focus on a narrow set of molecular targets that are suspected to play a role in cancer development and progression.Targeted therapies are selected or engineered to interact with their target, while many mainstream chemotherapies were discovered because they kill cells.

Targeted therapies are frequently cytostatic (that is, they prevent the proliferation of cancerous cells), whereas conventional chemotherapy medicines are cytotoxic (that is, they kill tumor cells). Combinations of metal clusters and organic linkers fabric the structure of MOFs, which gives them many desirable properties (including a large surface area and pore volume, surface chemistry, and a tunable pore environment) and allows for their application in a wide range of imaging and drug delivery systems. Biocompatibility, large drug payloads, and the ability to hybridize with a wide range of functions make MOFs an attractive option for targeted drug delivery [8,50]. For biomedical applications, nanoscale MOFs that are Zr-linked, such as MOF-808, offer some significant benefits in particular [14]. To improve chemotherapy’s therapeutic efficiency and achieve a highly selective target in cancer cells, nanoparticles loaded with floxuridine and carboplatin and further functionalized with a poly (acrylic acid-mannose acrylamide) glycopolymer coating was developed. Specifically, in HepG2 human hepatocellular carcinoma cells, the modification boosted the absorption of the nanoparticles and provided a significant selective drug delivery with great cytotoxicity. These findings demonstrate that MOF-808 is a promising choice for future drug delivery investigations [14].

## 3. Synthesis, Functionalization, and Modification of MOF Nanomaterials for Targeted Cancer Drug Delivery

### 3.1. Direct Assembly Technique

In addition to being directly encapsulated, some cargo molecules or their prodrug form can be utilized as ligands to actively contribute to the development of framework structures through coordination bonds between the cargo’s accessible coordinated functions and particular metal nodes [51]. A few chemotherapy agents, including pamidronate, zoledronate, methotrexate, and several platinum-based anticancer agents, as well as photoactive cargos for phototherapy, have been effectively introduced into MOFs. Through the gradual breakdown of these cargo or prodrug ligands into active components in a physiological milieu, this type of MOF nanoparticle can achieve its entire therapeutic function. The most significant aspects of this technique are its uniform cargo distribution and increased loading within the NMOF matrix, but it is important to completely evaluate how to preserve the therapeutic action of those cargo ligands during the synthesis process [52,53].

Coordination modulation is currently widely used in MOF chemistry due to the exquisite control it provides for the synthetic chemist. The technique evolved from early attempts to regulate the particle size of MOFs into a set of diverse synthetic protocols to manufacture single crystals, simplify synthesis, induce defects, and regulate a variety of physical properties, and, additionally, it has recently provided conditions in complicated delivery systems.

Related methods are currently being developed, such as multivariate modulation to increase the storage of numerous cargo molecules and pore complexity from the MOFs porosity, which allows for the simultaneous control of surface chemistry and particle size. Coordination modulation is currently motivating other techniques to exercise kinetic control while applying to substitute materials because the capacity to influence self-assembly kinetics has the potential to be highly strong. Even in previously well-researched chemical regions, the kinetic control provided by the various modulation strategies should aid in the identification of novel materials [54].

The path is taken by self-assembly to create nanocomposites containing upconversion nanoparticles that are homogeneously paved over MOFs. This approach, which is primarily driven by electrostatic interactions, can be utilized to combine various upconversion NPs with various MOFs. The as-synthesized composites are helpful for applications such as luminescence-monitored medication delivery. They can also be utilized to create composites with distinctive architectures, such as MOF@upconversion NPs@MOF sandwiched nanocomposites (Figure 4) [55].

To create novel functionalized heterogeneous catalysts of Cage@FDU-ED during a reaction of (2,2,6,6-tetramethylpiperidin-1-yl)oxyl, metal-organic cages were placed inside mesoporous carbon with amino functions. The discovered bifunctional catalyst has an improved catalytic activity, selectivity, and recyclability due to the orthogonal properties of the segregated actively catalyzing locations, resulting in an overall transformation yield of up to a 96% conversion. It is possible to achieve a constant and highly effective chemical transformation by carefully designing the catalytic sites in both the mesoporous matrix and MOF cages separately (Figure 4) [56].

### 3.2. Encapsulation Technique

Active compounds, primarily anticancer medicines, have been effectively integrated into nanoMOFs utilizing the three primary methodologies depicted in Figure 5. This section will focus on the specifics of how biomolecules are enclosed within the pores of MOFs. Utilizing the high and modifiable porous structure of MOFs and encapsulating bioactive molecules within the pores of MOFs can be a simple and effective way to overcome the limitations of the surface attachment method. However, as the majority of biomacromolecules have diameters >2 nm, the greatest difficulty in capturing biomolecules comes in the fabrication of MOFs with large pore spaces. The International Union of Pure and Applied Chemistry (IUPAC) divides nanocomposites into three main categories determined by the size of their pores: macroporous (>50 nm), mesoporous (2–50 nm), and microporous (2 nm). Mesoporous MOFs are widely used as host matrices for biomolecules due to their ability to shield them from environmental perturbations such as pH and temperature shifts and organic solvents. In addition, the microenvironment around the encapsulated biomolecules can be changed by precisely controlling the structure or property (e.g., functionality, charge, or lipophilicity) of the pore walls of MOFs, creating optimal conditions for biomolecules’ activities or applications [57]. Particularly attractive for drug entrapment are Materials Institute Lavoisier (MIL) MOFs constructed from centers of trivalent metal and bridging ligands of polycarboxylic acid, which create extraordinary surface areas ranging from 1500 to 5900 m^2^/g and huge pore diameters ranging from 25 to 34 Å [58]. MIL MOFs have been successfully loaded with a wide variety of different sorts of active compounds, including anti-inflammatory, anticancer drugs, metallodrugs, antiviral, nitric oxide, and peptides (Table 2). Several additional MOFs, including those based on zinc, copper, and zirconium, were utilized as drug carriers. Due to the possible advantages that metal ions such as Gd, In, and Ni could impart, several MOFs based on them were also examined for drug encapsulation, and their imaging characteristics were also investigated. On the other hand, it has been noted that the toxicity of metals might be an obstacle to the use of certain MOFs in medical applications.

### 3.3. Post-Synthesis Technique

In the absence of functional groups, the structurally unmodified form of an MOF may restrict its usefulness. Toward this aim, post-synthetic modification (PSM) is performed to increase the functional groups attached to MOFs (Figure 5) [109,110], thereby extending their potential spectrum of applications. Briefly, PSM is a systematic approach to surface functionalization that is used to introduce functional groups into MOFs [111,112]. If suitable functional groups are installed, PSM has the potential to enhance the chemical and physical characteristics of materials. This alteration also serves to govern the overall utilization (colloidal stability or self-assembly under varying conditions) and its interactions with its surroundings, such as a target-specific accumulation [110]. Surface functionalization has many additional benefits, including (i) preventing nanoparticle aggregation; (ii) phase transfer, transferring nanoparticles from one solvent to another solvent (e.g., organic solvent to water); (iii) allowing nanomaterials to associate with particular biomolecules of attention, such as nucleic acid, in delivery, therapeutic use, and biological networks for imaging; and (iv) modification using fluorescent dyes for functionality [113].

### 3.4. In Situ Synthesis Technique

Given their host–guest characteristics and ease of chemical synthesis modification, one of the most important and potentially useful applications of MOFs is the delivery of drugs. The inorganic segment of MOFs regulates medication release, while the organic segment of MOFs can be customized to encapsulate a variety of pharmaceuticals. Even though these substances have demonstrated an adequate drug loading capacity and controllable drug distribution behavior, few studies on drug delivery in MOFs have been established up until this point. A nanoscale MOF was recently used for effective medication loading and delivery [114].

## 4. Applications of MOF Nanomaterials in Targeting Cancer Therapy

### 4.1. Active Targeted Cancer Therapy by MOF Nanomaterials

Using an active targeting method, nanoparticles (NPs) can increase the intracellular concentration of medications in malignant cells while minimizing damage in healthy cells. Bioscience-enhanced NPs are actively being developed for targeted drug administration, biomarkers for cancer using bimolecular profiling, and in vivo tumor imaging. The patient will need to take fewer doses more frequently, the drug will have a more consistent impact, there will be fewer side effects, and the levels of the drug in the blood will fluctuate less. These are all benefits of the active targeted release system. Active targeting entails adding various ligands to the medication or DDS, including vitamins, peptides, antibodies, sugars, and biological proteins. These ligands interact with cell receptors to form complexes that induce the drug to assemble within the target cells [115,116,117].

A methodical approach to creating an MOF that is two-photon active is through a click reaction of PCN-58-Ps. Hyaluronic acid is additionally added to PCN-58-Ps by coordination to give it cancer-cell-specific targeting characteristics. The improved composite of PCN-58-Ps-HA consequently demonstrates the strong activity of two-photon (up activation by a laser with a 910 nm wavelength) and light-activated ROS of 1O_2_ and O_2_^•−^ production capabilities (Figure 6a,b). Future clinical applications of deep-tissue cancer imaging using two-photon PDT that is activated by NIR light and treatment have a large amount of potential due to the interaction of these two crucial variables inside the framework of PCN-58-Ps-HA [118].

Glycyrrhetinic acid (GA), lactobionic acid (LA), dual ligands of GA and LA, and folic acid (FA) were designed and built as effective multifunctional DDSs for combating hepatocellular carcinoma (HCC). Doxorubicin was loaded into the Zr (IV)-based NMOF (NH_2_-UiO-66) nanoscale. It was established that pure NH_2_-UiO-66 was safe when used with HSF cells based on biocompatibility experiments; however, DOX-loaded NMOF was discovered to be more harmful to HepG2 cells by flow cytometry (Figure 6c). The created dual-ligated NMOF demonstrated a pH change in response to the DOX release [119].

### 4.2. Passive Targeted Cancer Therapy by MOF Nanomaterials

To achieve screening and therapeutic uses in cancer nanobiotechnology in vivo, nanoparticles must be transported to cancer locations. To achieve this, two broad strategies have been employed: passive targeting and active targeting (Figure 7) [120,121,122]. Passive targeting exploits the biological characteristics of tumors to enable nanocarrier accumulation in tumors via an improved permeability and retention (EPR) [123]. For the accumulation of NPs in tumors, passive targeting relies on aberrant gap junctions (100–600 nm) in the endothelium of tumor blood arteries. MOFs can typically be adequately manipulated at the nanoscale for passive targeting. Park and colleagues explored the HeLa human cervical tumor cell absorption of a porphyrinic MOF (PCN-224) by altering the particle size to boost the cytotoxicity and internalization via passive targeting [124]. They found evidence that MOFs improve the photodynamic performance. The cytotoxicity of photosensitizers is minimal when MOFs are not included. The use of TCPP@PCN-24 in photodynamic therapy was found to be most successful when the particle size was 90 nm, whereas the use of this combination was found to be least effective when the particle size was 190 nm. Due to its superior retention impact in the tumor area, Duan also showed that particles of size 60 nm AZIF-8 have an anti-tumor effect superior to those of other sizes [125].

### 4.3. Physicochemical Targeting Cancer Therapy by MOF Nanomaterials

#### 4.3.1. Light-Responsive Targeted Cancer Therapy by MOF Nanomaterials

Recent advances in light-mediated nanomedicines, particularly their minimally invasive abilities and great spatiotemporal accuracy, have made them attractive methods for precisely controlling the therapeutic activation and imaging probes both in vivo and in vitro (Figure 8) [126,127], particularly the light-activated MOF-based therapeutic system, which not only offers imaging-guided or combination therapies but also improves the laser penetration depth and targeting [128,129].

A biodegradable and biocompatible MOF was created for effective drug loading and controlled release by developing Au (gold nanorods) @ZIF-8 (crystalline zeolitic imidazolate framework-8). A significant drug loading efficiency of roughly 37% was achieved by the Au@ZIF-8 while loading doxorubicin. The ZIF-8 layer was swiftly destroyed by NIR light or a mildly acidic environment, which led to an on-demand medication release at the tumor location. More significantly, because of the synergistic effects of photothermal treatments and chemotherapy, highly efficient cancer treatment was accomplished in both in vitro cell experiments and in vivo tumor-generating naked mice experiments under the irradiation of a near-infrared laser. The in vivo study also demonstrated Au@ZIF-8’s good biocompatibility (Figure 8) [130].

The nanoparticle is made up of a cancer cell membrane shell and a metal–organic framework core coated in MnO_2_ nanosheets (CM-MMNPs). The H_2_O_2_ and H^+^ responsiveness of the MnO_2_ layer allows it to generate O^2^, enhancing the generation of O_2_-mediated singlet oxygen (1O_2_) for photodynamic therapy (PDT). Additionally, the resultant Mn^2+^ is a superior MRI contrast material. The CM-MMNPs are given cellular endocytosis that occurs with strong stability and integrity and a dependable homologous cell-targeting capability by the addition of membrane proteins and cell membranes. This multifunctional nanoparticle offers a new paradigm for targeted therapy, diagnosis, and treatment, and can treat cancer cells’ hypoxia with PDT [127].

#### 4.3.2. pH-Responsive Targeted Cancer Therapy by MOF Nanomaterials

Chemodynamic therapy (CDT), a new therapeutic, is described as the generation of cytotoxic ^•^OH at tumor locations via a Fenton or Fenton-like reaction [131,132]. Because CDT relies on the Fenton-type process, which primarily catalyzes endogenous hydrogen peroxide (H_2_O_2_) to produce ^•^OH, the low H_2_O_2_ level in solid tumors will limit its usefulness. Therefore, combining GOx-induced fasting therapy with CDT is a brilliant move that has the potential to greatly boost synergistic therapeutic effects. A pH-responsive nanoplatform was developed by encapsulating GOx and natural hemoglobin (HB) together in ZIF-8 via co-precipitation to combine CDT and starvation therapy effectively [133,134]. Another study revealed the creation of a biodegradable mesoporous Fe (III) polycarboxylate MOF exhibiting the pH-sensitive and reversible aggregating ability to specifically target the pulmonary for drug administration in order to efficiently suppress lung cancers (Figure 9) [135]. Using a lung metastasis model, the nanoparticles were randomly aggregated in the capillaries and subsequently disaggregated after 24 h, allowing the encapsulated medication to be released. The pH-responsive property might not just facilitate the launching of medications in the appropriate location within tumor cells but may also be used to synthesize MOFs that disintegrate at a specific pH, resulting in rapid drug release [136].

#### 4.3.3. Magnetic-Field-Responsive Targeted Cancer Therapy by MOF Nanomaterials

A hybrid magnetic nanocomposite with a well-controlled size distribution, typically 100 nm, is produced by combining MIL-88B-NH_2_ MOF-structured Fe_3_O_4_ magnetic NPs using tailored synthetic media, including F127 copolymer as a stabilizing agent and acetic acid as a modifying agent. The nano MOF has also been used as a nanocarrier for controlled medication release on demand and effective drug delivery. Carmustine and Mertansine, two glioblastoma medications, were perfectly inserted into the MOF structure’s pores to reduce their potent internal harmful effects, which restrict their clinical use. Moreover, localized heating was caused as a result of the magnetic local minimum included in the MOF NPs when a modified magnetic field was practiced on the magnetic nanocomposites loaded with DM1 to accomplish controlled drug release. To verify the therapeutic effectiveness of the DDS on U251 glioblastoma cells, in vitro cytotoxicity experiments were performed (Figure 10) [137].

An Fe_2_Mn(3-O) cluster was used to create an FeMn-based ferromagnetic MOF. FeMn-MIL-88B acquired its ferromagnetic properties with the addition of Mn. As a model drug, 5-Fluoruracil (5-FU) was entrapped in MOFs, and its regulated release that was responsive to both pH and H2S stimuli was achieved. In tumor microenvironment (TME) simulation media, FeMn-MIL-88B revealed an impressive capacity for loading 5-FU (43.8 wt%) and the quick release of the drug. Additionally, the carrier’s cytotoxicity profile against embryonic kidney cells of humans indicates no negative effects (100 g/mL). The low toxicity values (LD50; Mn = 1.5 g/kg, Fe = 30 g/kg, and terephthalic acid = 5 g/kg) of the MOF’s basic components can be attributed to the less hazardous effect on the cell viability (Figure 10) [138].

#### 4.3.4. Redox-Responsive and Targeted Cancer Therapy by MOF Nanomaterials

The abnormal physiological properties of cancerous tissue are reflected in the presence of distinct cellular microenvironments, such as reducing, acidic, and enzyme environments [139]. With significant reducing capacities, the reduction states and oxidation of nicotinamide-adenine dinucleotide phosphate (NADPH) and glutathione (GSH) allowed them to primarily manage the reducing environment of cancer cells [140]. In a reducing environment, GSH that has a concentration greater than that of NADPH plays a crucial function in microenvironment regulation. GSH fragments and forms disulfide (S-S) links to regulate the cellular reducing environment. The intracellular tumor GSH concentration was higher than the extracellular tumor GSH concentration. Tumor tissues had four times greater GSH concentration than normal tissues. To take advantage of these features, numerous nanoscale drug delivery devices for monitoring the reducing environments of cancerous tissue and triggering drug release by disulfide bond breakdown in GSH-responsive nanomaterials have been constructed (Figure 11) [139,141]. Disulfide bonds have several important applications in MOF fashion, including the architecture of ligands and the alteration of surfaces [142,143,144]. Zhao et al. described the fabrication of Mn-S-S, a glutathione-responsive MOF system, using Mn^2+^ and dithiodiglycolic acid as ligands, thus inserting the S-S link into the MOF ligand [143]. The disulfide bond was cleaved in the existence of glutathione, which allowed for the medication (DOX) to be successfully released from its encapsulation. In addition, the Mn^2+^ in Mn-SS@MOF showed an improved T1 contrast in the medical diagnostic of magnetic resonance imaging (MRI). In addition, Liu et al. designed an MOF that is both redox-sensitive and tumor targeting by attaching folic acid and functional S-S anhydride to the organic linkages of UiO-66-NH_2_ MOFs [145]. Drugs in MOFs are released in response to redox stimuli, and this is achieved by the overexpression of GSH in tumor cells, which causes an assault on the thiolate moiety and cleaves the S-S bonds.

#### 4.3.5. Thermosensitive MOFs for Targeted Cancer Therapy

An MOF contains organic linkers that are synthesized utilizing NbO-type Zn^2+^, which contains two structurally identical tetracarboxylate ligands with pyrazine or pyridine moieties. The trivalent europium ion (Eu^3+^) and the pyridinium hemicyanine dye 4-p-(dimethylamino)styryl]-1-methylpyridinium (DSM), both of which have cationic red-emitting units, were embedded in various composites, and their potential as ratiometric temperature probes was assessed. These dual-emitting composites’ temperature-responsive luminescence was examined, along with their characteristic features of temperature resolution, relative sensitivity, spectral repeatability, and luminous color change. The Eu^3+^@ZnPZDDI and Eu^3+^@ZJU-56 exhibit good sensor temperature ranges and high relative sensitivities, indicating that the composites can be extensively designed by integrating the guest and host units (Figure 12) [146].

By post-synthetic modifying, a copolymer comprising N-isopropyl acrylamide and acrylic acid was utilized to cover the MOF. The additional molecules can be released in an “on-off” fashion thanks to the polymer’s quickness as well as the transition from a reversible coil to a globule, which is pH- and temperature-responsive. When the polymer assumes a coil shape at low temperatures (25 °C) or a high pH of 6.86, the additional molecules are quickly freed from the MOF. The release of the attached molecules is blocked when the polymer takes on the shape of a globule with a pH of 4.01 and/or warm temperatures of over 40 °C. Even once the release has begun, it can be stopped by adding external stimulation (Figure 12) [147].

### 4.4. MOF-Based Bionic Immune for Targeted Cancer Therapy

The immune system is a sophisticated biological architecture crucial for recognizing and removing invading invaders, destroying abnormal cells, and preventing the development of tumors [148,149,150]. Therefore, immune cell therapy presents significant opportunities for treating diseases such as cancer, autoimmune diseases, inflammation, and infections. There are many different kinds of immune cells, each of which performs a specific function and has the potential to be used as a live treatment for a variety of disorders [151]. To optimize pharmacokinetics and degradation, a bionic nanoplatform can deliver drugs to the immune system (Figure 13). These techniques have strengthened drug bioavailability by providing extra protection and targeting, encouraging the enhancement and evolution of bionic solutions [152,153,154]. In expanding the variety of medications that can be loaded, Gong et al. came up with the innovative idea of developing a hybrid coating made up of macrophage and tumor-cell membranes. This coating combined the characteristics of both kinds of cells [155]. These cells targeted particular homogeneousness and metastasis, and also aggregated in inflammatory areas. Thus, this combination offers promise for nanobionic architecture. However, the bionic system of cancer cell membranes needs to be improved and made more open to innovation.

### 4.5. MOF-Based Nanotherapeutics as Gene Delivery for Targeted Cancer Therapy

The genetic advancements cleared the door for the introduction of genetic engineering, and, since 1980, gene therapy has become an increasingly prominent topic in cancer research. The three primary categories of gene therapy approaches are immunomodulatory, corrective, and cytoreductive [156]. Immunomodulation is the process of boosting the body’s immune barrier to effectively identify and destroy cancer cells. An important goal of corrective gene therapy is to restore the normal function of a gene whose mutation contributes to the development of cancer. Suicide gene therapy, in which a gene is inserted into cells that codes for an enzyme that converts a harmless prodrug into its toxic metabolite, is also being studied extensively as a means of treating cancer. It has been hypothesized that nucleic acids incorporated into MOF nanocarriers might be stabilized against degradation and taken up by cells more quickly. In addition, the steric and electrostatic impediment to aggregation could be enhanced by the surface configuration of an MOF with nucleic acids, resulting in an improved colloidal consistency. MOFs have found widespread application in the administration of drugs, the transport of siRNA, and the encapsulation of DNA/RNA, proteins, and polysaccharides, as well as in prokaryotic and eukaryotic organisms [157,158]. As an example, ICG@ZIF-8 was used by Liu et al. as a means of electrostatically adsorbing siRNA (ICG@ZIF-8@siRNA) to promote siRNA diffusion under laser control [159].

### 4.6. MOF Multi-Targeted Response for Cancer Therapy

With self-amplified releasing and improved penetrating, a pH and ROS dual-sensitive biodegradable MOF nanoreactor-based nanomaterial was created to provide GOx and 1-MT together for combination oxidation/starvation treatment and IDO-blockade malignancies immunotherapy (Figure 14a–c). The comprehensive in vivo and in vitro results validated PCP-Mn-DTA@GOx@1-MT nanomaterial to not only proficiently strengthen immune system activation with a decreased sensitivity by GOx-activated starvation/oxidation intervention and IDO-blockade immunotherapeutic, but to also strategically overwhelm biobarriers and increase the distribution efficiency through the mildly acidic tumor cells [160,161,162].

As a novel method for cancer therapeutic applications, a nanoscale MOF platform will merge magnetic resonance imaging, photothermal therapy, and spatiotemporally programmable NO delivery. The MOFs are generated as a proof of concept using biodegradable Mn-porphyrin and Zr^4+^ ions as linking ligands. Mn-porphyrin gives the NMOF a robust T1-weighted MR contrast performance and a significant photothermal transformation for effective PTT through the incorporation of paramagnetic Mn ions into porphyrin rings. For heat-sensitive NO production, S-Nitrosothiol (SNO) is attached to the NMOF surfaces. Additionally, a single NIR (near-infrared) light triggers both the PTT and regulated NO release concurrently for their effective synergistic therapy in a single step. The tumor-bearing mice’s MR images reveal that NMOF-SNO exhibits effective tumor accumulation after intravenous injection. Tumors in mice given NMOF-SNO injections are fully suppressed when applied by NIR laser, demonstrating the effectiveness of the drug [163].

## 5. Challenge of MOF Nanomaterials in Cancer Treatment

### 5.1. Toxicity and Biocompatibility

For application in biomedical and pharmaceutical applications, MOFs must present toxicologically compatible characteristics. The cytotoxicity of several representative MOFs was evaluated on zebrafish embryos. At 120 h after fertilization, it was discovered that the viability of embryos that had been subjected to Co and Mg-MOF-74, UiO-66 and 67, and MIL-100 and 101 was not significantly different from those without MOFs (the control group), even at a concentration as high as 200 μM. On the contrary, the embryo viability rate (EV) was significantly reduced by ZIF-7 and 8 and HKUST-1. At a concentration of 200 μM, ZIF-7 was marginally harmful, with EV= 79.2%, whereas ZIF-8 was more toxic, with EV = 33.3%. NanoHKUST-1 was extremely hazardous even at a concentration of 20 μM (EV = 0%). It was highlighted that the absorption of solubilized metal ions had a crucial role in determining the toxicity potential of MOFs [164]. Rats were used in the experiment to evaluate both the toxicity of trimesic acid and iron trimesic MIL-100 (Fe). MIL-100 and trimesic acid were administered intravenously at doses of approximately 220 mg/kg and 78 mg/kg, respectively, and the study revealed that weight growth and the animal behavior of treated rats were completely normal when matched with the group that served as the control. This led to the conclusion that the concerned ingredient is suitable for drug delivery [165].

The biocompatibility testing of MOF building blocks is necessary before MOFs can be used in biomedical applications because some forms may have a harmful influence due to the degradation of MOFs within cells [166]. Additionally, it is also essential to examine the biocompatibility of MOFs with different types of cells since the outcomes may differ. In vitro analyses were conducted to study the biocompatibility of three distinct metal systems of MIL-100 MOFs (Fe, Al, and Cr). Even at high doses, they did not produce in vitro cell toxicity in the liver (HepG2) and lung (Calu-3 and A549) cell lines. Only the toxicity of MIL-100(Fe) was seen in the Hep3B cell line [167]. However, the other research, observing the biocompatibility of ZIF-8 concerning six distinct cell cultures, each one representing a different part of the human body (skin, breast, blood, kidneys, bones, and connective tissues), showed that ZIF-8 at concentrations higher than the threshold level of 30 μg/mL exhibited cytotoxicity [168]. This might be related to the influence of liberated Zn^2+^ on the generation of mitochondrial reactive oxygen species. Despite the tremendous amount of research conducted in the field of nanomedicine and enhanced nanocarrier therapeutics directly attacking cancer, the process of bringing medication from the research lab to the industry is frequently intermittent and highly slow. Most of the issues can be traced back to the lack of rigorous monitoring of the safety and effectiveness-by-design strategy for nanomedicines.

### 5.2. Drug Release before Reaching the Target Cancer

The performance and safety of MOFs are significantly impacted by their chemical stability. The stability of MOFs is affected by several variables, including pH, temperature, humidity, solvents, metal ions, and biological molecules. Different stability levels may be needed depending on the application. For instance, MOFs must be able to survive the stomach’s acidity and the intestine’s alkalinity when given orally. MOFs must be resistant to metal ions and tear fluid for ocular administration [169]. The stability of MOFs for drug delivery has been improved through the development of several methods: the addition of protective layers such as polymers; the selection of more stable metal nodes or organic ligands; and the addition of functional groups or linkers to MOF structures or compositions [170]. The porosity, loading capacity, release kinetics, biocompatibility, and toxicity of MOFs may also be impacted by these tactics. Hence, for each application, a careful balance between stability and functionality needs to be struck [171].

Because of their modifiable surface, tunable size, high active ingredient loading, good biocompatibility, and, most importantly, their capacity to be selectively distributed in tumor cells via an increased retention and permeability, MOFs made up of bridging ligands and metal attachment sites were investigated as a new innovative system for the enhanced treatment and diagnosis of cancer. However, the following points need to be studied more deeply. However, controlling drug release during cancer cell targeting is quite important.

The body typically experiences severe adverse responses as a result of chemotherapy medications. Therefore, it is critical to increase the targeted drug delivery system’s stability to prevent both medication leakage outside of the tumor and significant detrimental effects on the body. Cell-membrane-coated biomimetic approaches are already common; however, they have more expensive material needs. The development of an MOF-based DDS suited for clinical applications, from synthesis to in vivo process monitoring, and quality control still needs to be carried out [116].

### 5.3. In Vivo Studies and Applications

Despite the unparalleled benefits, more focus needs to be placed on in vivo investigations of MOFs, such as the toxicity and biocompatibility. More information is required to fully comprehend the metabolic and mechanical operations of MOFs in the body since they degrade. Numerous research studies on the cytotoxicity of MOFs in vitro have been recently published. However, cell models do not reveal the same biocompatibility of MOFs in the body, despite the good biosafety at a given dose. The majority of research on the toxicity and metabolism of MOFs-based DDS have been limited to their anti-tumor activities in experimental animals.

The Fe_3_O_4_@C@PMOF metabolism in nude mice was studied in breast cancer nude mice. Fluorescent in vivo imaging with the nanoparticles is possible. The presence of fluorescent dots in the lymph and liver nodes proved that NPs could take part in both lymphatic and blood circulation. The tumor region then expanded to form the tissue with the maximum fluorescence intensity. The NPs were expelled from the body through stool after 8 days. The mice who received the injections acted normally, and their weight did not significantly drop. The primary organs of mice did not exhibit any pathological changes eight days after injection, and there was good biocompatibility [172].

The toxicity of a porphyrinic MOF nanosystem to the major organs, blood, and tissues of mice was assessed as part of a study on in vivo biosafety. All indications were in the usual range, proving the nanosystem’s high level of safety [173,174]. Iron (III) MOFs’ in vivo toxicity was examined using markers such as serum, enzymes, and histology, all of which were consistent with low toxic effects. The liver and spleen isolate the nanomembrane, which is further biodegraded into carboxylic acids and iron and subsequently eliminated directly in the feces or urine while still preserving iron homeostasis. This demonstrates that iron (III) carboxylate MOF NPs are non-toxic and biodegradable [175].

### 5.4. Quality Control from Laboratory Scale to Industrial Scale

Research on MOF’s biomedical efficiency is still being conducted in small-scale production and laboratory testing at this time. When MOFs are produced in large quantities, it can be challenging to control their quality, which can cause fluctuations in the pore size and material size. Additionally impacted are the drug loading capacity and release rate. Therefore, one of the most difficult issues is the construction of stable and manageable MOFs (Table 3).

## 6. Future Perspectives of MOF Nanomaterials in Cancer Treatment

Although the biofunctionalization of MOFs has been the subject of substantial investigation in the area of nanomedicine basic research, its use in clinical therapy still has a long way to go. MOFs with good biocompatibility and non-toxicity or low toxicity require optimizing the preparation and synthesis of the nanomaterials such that they can circulate for an extended period and are efficiently eliminated via metabolism. An additional experiment is necessary to fully understand the biodegradability and stability of MOFs in the body. There is a requirement for reinforcement in light of the effects of MOFs nanomedicines on the physiological activities of organisms. Going forward, building dynamics on MOF-responsive biomaterials and responding to specific physiological stimuli by the targeted and timed release of bioactive molecules have significant research significance and application potential.

## 7. Conclusions

Biomolecule–metal–organic framework composites have advanced substantially over the past decade to become novel platforms for an extensive variety of solutions in the identification and treatment of cancer. Here, we reviewed recent improvements in the fabrication of biomolecule–MOF composites and their application in cancer treatment. Integrating biomolecules into MOFs has been accomplished using a wide range of different approaches, including surface adhesion, covalent bonding, pore encapsulation, in situ synthesis, and bioengineering MOFs, and has led to significant progress in the development of targeted therapy for cancer. Due to the advent of MOFs, it is now possible to overcome the limitations of conventional cancer therapeutic approaches, such as vulnerability, inadequate absorption, low aqueous solubility, low selectivity, high lethality, and multidrug resistance. Although MOFs are gaining importance for cancer prevention and treatment, each inclusion approach has advantages and disadvantages. Their promise as medication carriers in cancer treatment is still in its infancy. In fact, a significant number of MOFs are now going through clinical or preclinical testing in preparation for being approved and made accessible for purchase. Global research on using nanostructured frameworks for the goal of generating patient-specific drug delivery systems will assist the synthesis and development of MOF–drug composites for real-world applications. With the assistance of future development strategies of MOF-based materials for cancer therapy, it will be possible to reduce the cost of cancer treatment while simultaneously increasing the patient survival and quality of life.

## Figures and Tables

**Figure 1 pharmaceutics-15-00931-f001:**
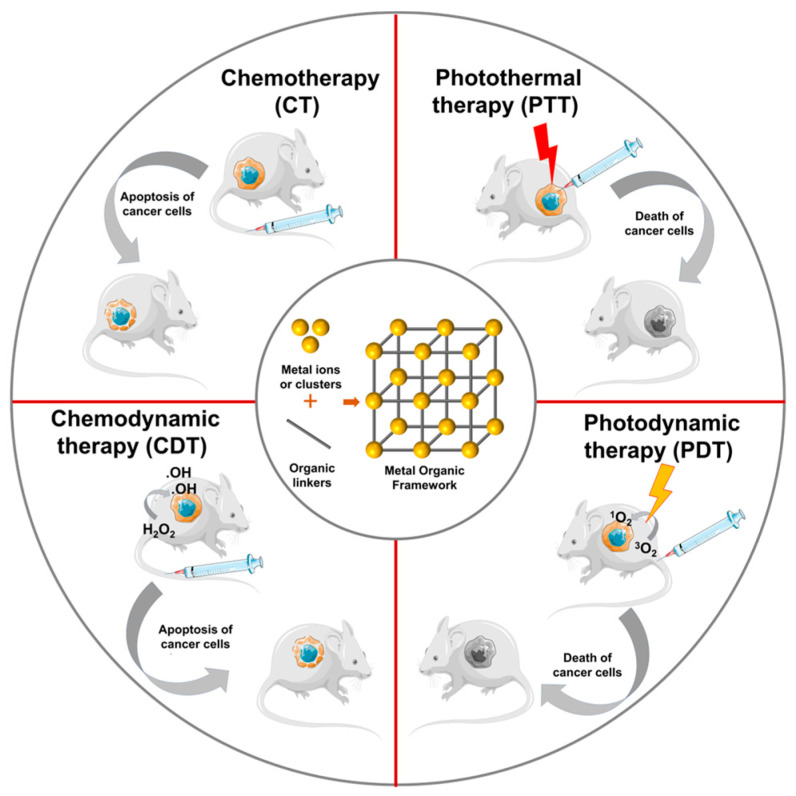
Some monotherapies are based on MOFs. Metal ions and an organic ligand are the two components that make up MOFs. The organic ligand joins the metal ions to form larger arrays. MOFs have several desirable structural properties that make them excellent candidates in the fields of drug administration and cancer therapy.

**Figure 2 pharmaceutics-15-00931-f002:**
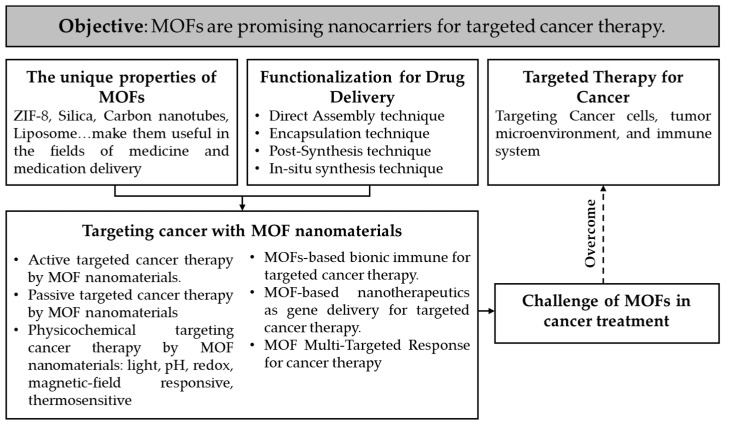
The logical structure of the review.

**Figure 3 pharmaceutics-15-00931-f003:**
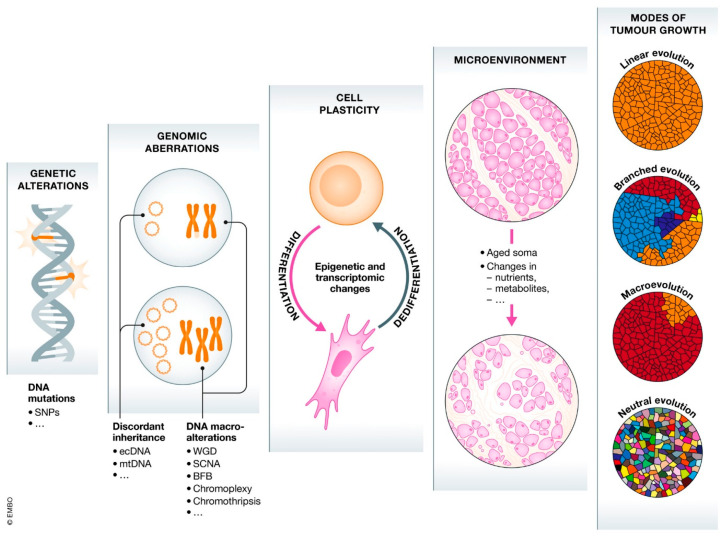
The gradient of cancer development. Microscopic to macroscopic (left to right) examples of the several factors of tumor evolution. Reproduced with permission from [47]. Copyright 2021 The EMBO Journal.

**Figure 4 pharmaceutics-15-00931-f004:**
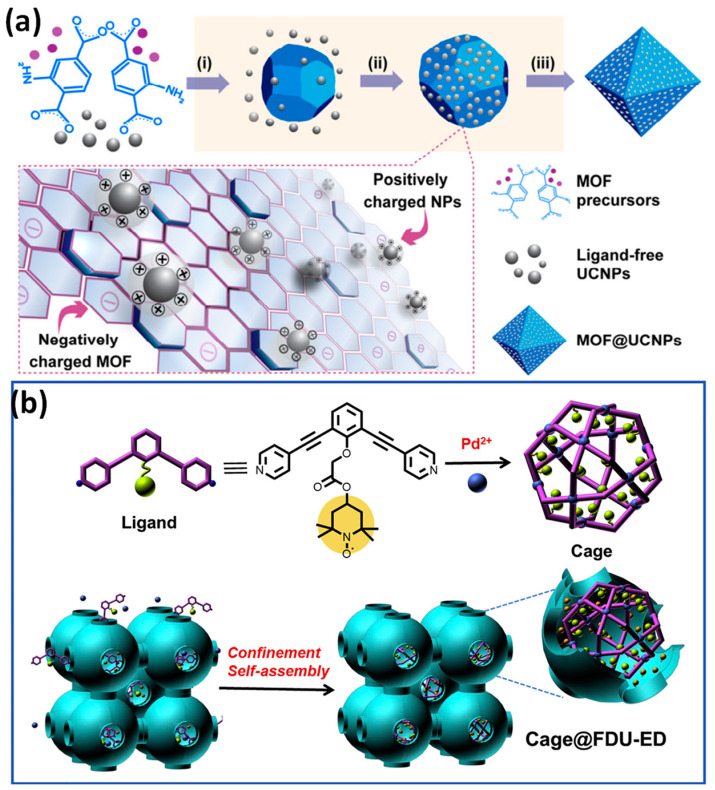
(**a**) Upconversion nanoparticles (UCNPs) and MOF nanomaterial manufacturing is shown schematically. Direct mixing results in the formation of nanocomposites from the reaction intermediates of MOF and ligand-free UCNPs. The processes demonstrate the three hypothesized formation mechanisms: (i) nucleation of MOFs, (ii) electrostatic bonding of NPs to MOFs, and (iii) production of nanocomposite materials; (**b**) catalyst with two functions for single-pot sequential reactions using confinement self-assembly and M12L24 cage C in FDU-ED cavities mesoporous to produce cage@FDU-ED is shown schematically. Reproduced with permission from references [55,56].

**Figure 5 pharmaceutics-15-00931-f005:**
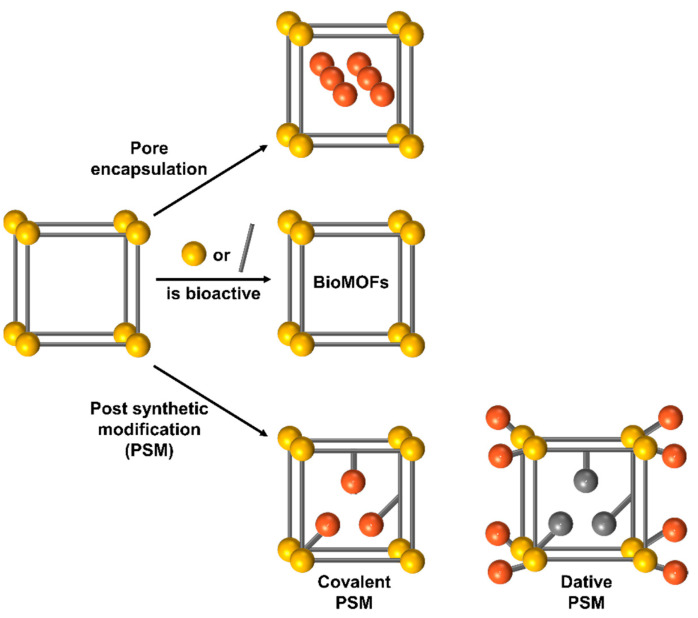
Combining drugs with hybrid metal–organic framework nanoparticles. The aforementioned three basic techniques have been successfully used to insert active compounds, particularly anticancer medicines, into the MOFs.

**Figure 6 pharmaceutics-15-00931-f006:**
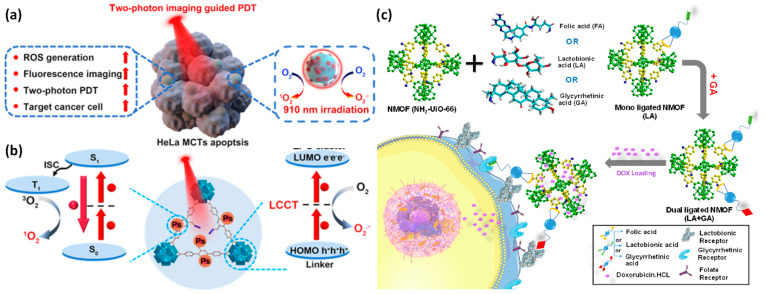
(**a**) Schematic illustration displaying the PDT guided by two-photon fluorescence imaging, the target cancer cell, and the light-induced ROS creation; (**b**) mechanisms of ROS created by PCN-58-Ps-HA by light irradiation of two-photon; (**c**) drug delivery systems using functionalized Zr-based NMFOs (NH_2_-UiO-66) for the targeted therapy of hepatocellular cancer are represented systematically. Reproduced with permission from references [118,119].

**Figure 7 pharmaceutics-15-00931-f007:**
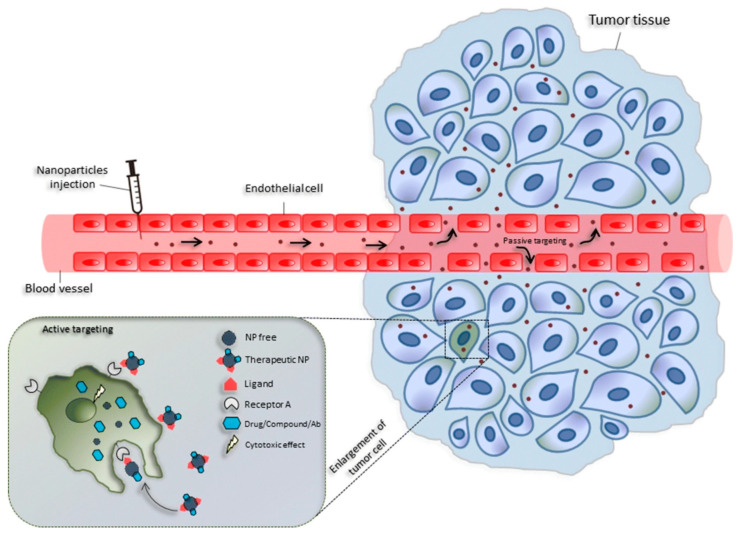
Cancer treatment involves passive and active targeting of nanoparticles. Extravasation of nanomaterials through enhanced permeability (EPR effect) of the tumor vasculature enables passive tumor targeting. By functionalizing nanomaterials aimed specifically at ligands that improve cell-specific identification and adherence, active tumor targeting (left inset) is possible. Reproduced with permission from [122]. Copyright 2018, Springer Nature.

**Figure 8 pharmaceutics-15-00931-f008:**
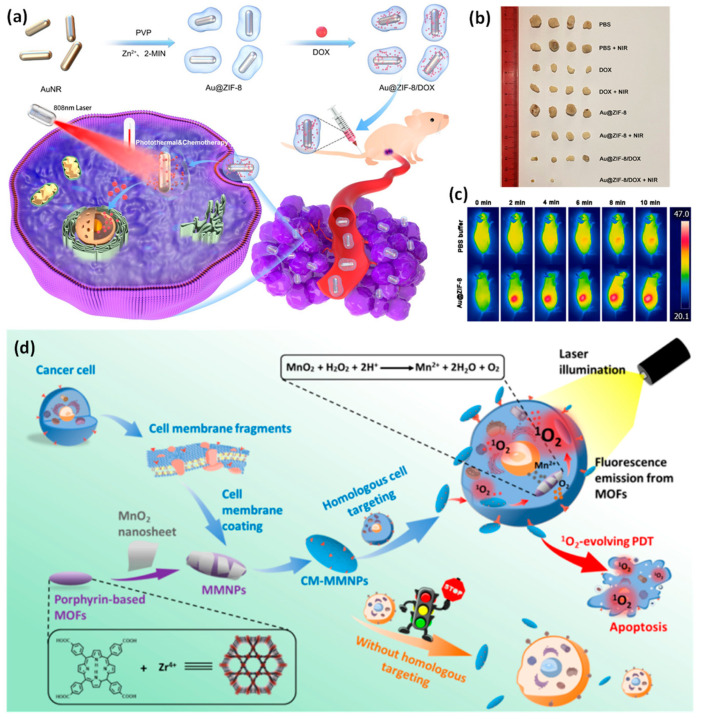
(**a**) A diagram of the core–shell Au@ZIF-8/DOX synthesis process for in vivo chemo-photothermal cancer treatment; (**b**) intravenous injections of Au@ZIF-8 or PBS solution followed by in vivo infrared thermal imaging of MCF-7 tumor-bearing mice following activation with a 1 W/cm^2^ 808 nm laser for 10 min; (**c**) a characteristic image of tumors removed from mice treated in each group; (**d**) the nanostructure is made up of a cancerous cells membrane layer and a metal–organic framework core coated in MnO_2_ nanosheets for MRI and fluorescence dual-mode imaging, homologous targeting, and photodynamic therapy for the diagnosis and treatment of cancer cells. Reproduced with permission from references [127,130].

**Figure 9 pharmaceutics-15-00931-f009:**
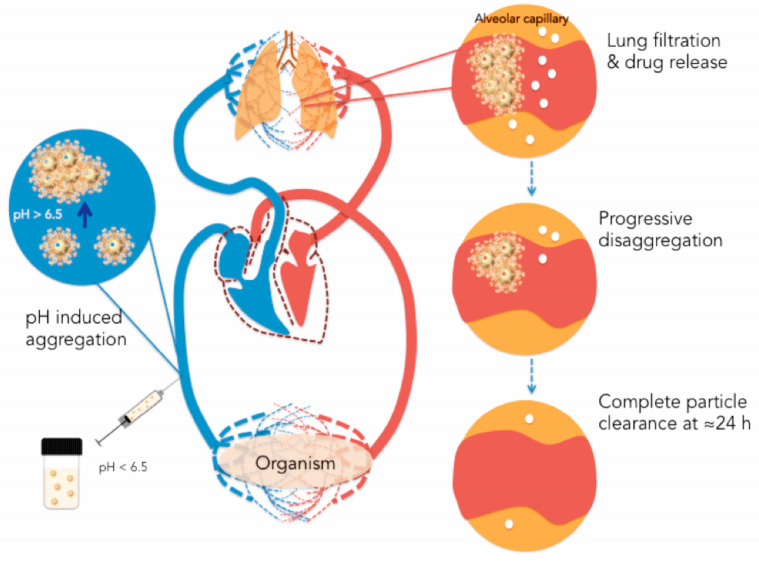
To target the lung, an iron-based biodegradable MOF that exhibits pH-responsive as well as reversible aggregating activity was created. Within twenty-four hours, the nanoparticles were enabled to autonomously aggregate at the pulmonary capillaries and then disaggregate again. Reproduced with permission [135]. Copyright 2021, Wiley-VCH Verlag GmbH & Co. KGaA, Weinheim.

**Figure 10 pharmaceutics-15-00931-f010:**
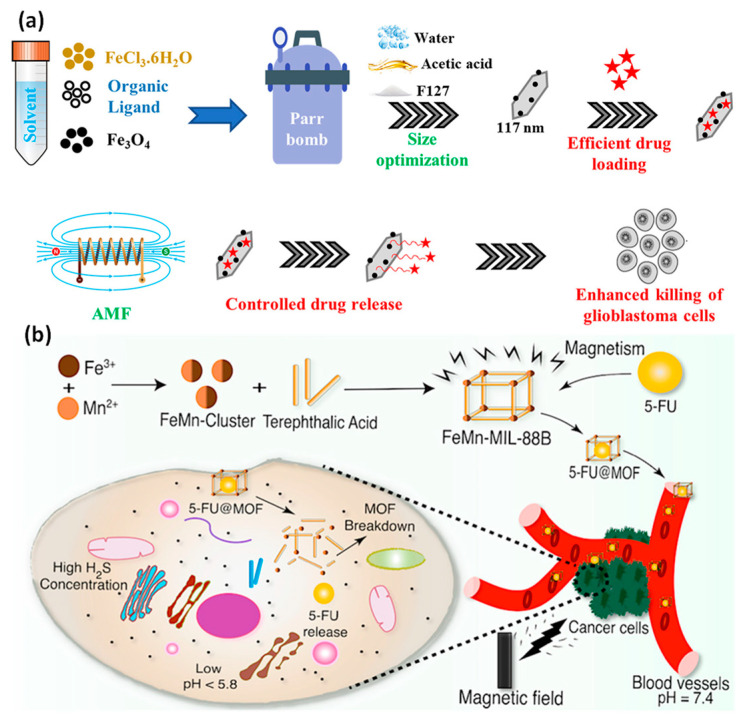
(**a**) F127 copolymer was used as a stabilizing substance during the synthesis of MOF Fe_3_O_4_ NPs with MIL-88B-NH2 structures, and these nanoparticles were used to deliver drugs using an alternating magnetic field and localized heating effect; (**b**) a preassembled Fe_2_Mn(3-O) cluster was used to create an FeMn-based ferromagnetic MOF that controlled drug release by dual stimulation, ferromagnetic nature, and low toxicity. Reproduced with permission from references [137,138].

**Figure 11 pharmaceutics-15-00931-f011:**
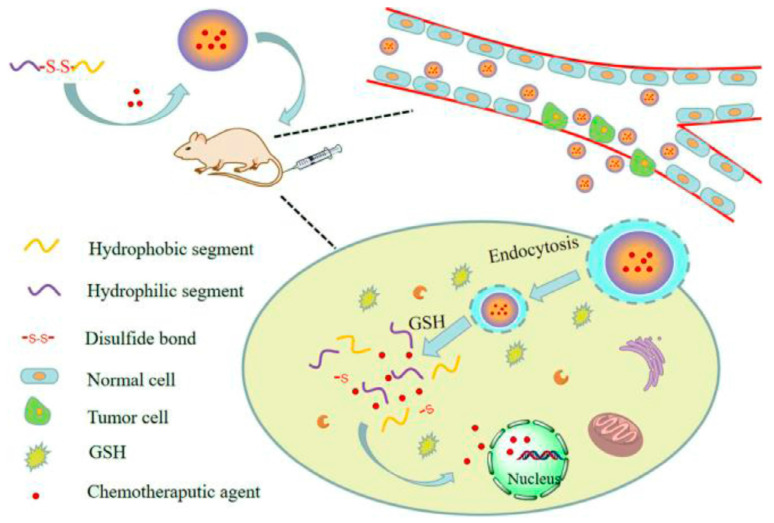
The biological process that is accountable for the release of redox-sensitive drug delivery devices. Reproduced with permission from [141]. Copyright 2019 Elsevier B.V.

**Figure 12 pharmaceutics-15-00931-f012:**
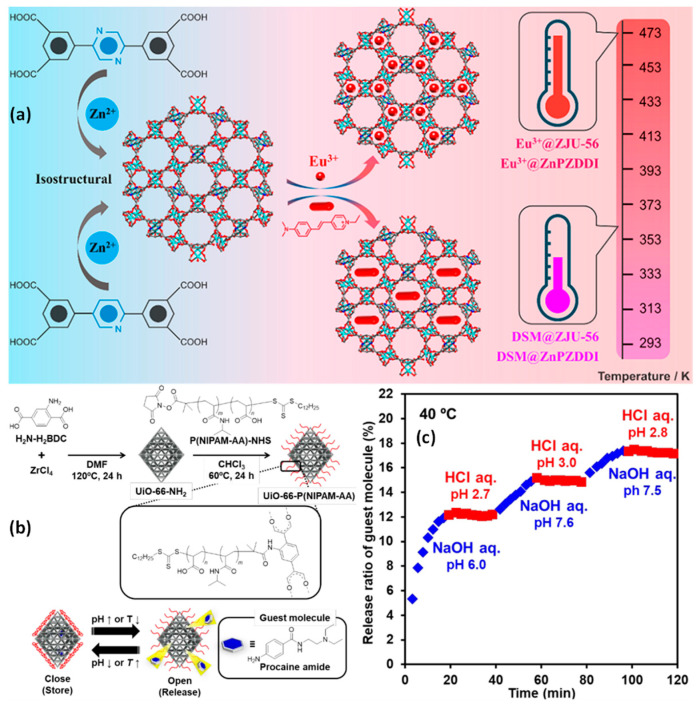
(**a**) Illustration of the fabrication of nanocomposites made from DSM@MOFs and Eu^3+^@MOFs for dual-emitting sensor nanomaterial; (**b**) schematic illustration for making an MOF that is tethered by P(NIPAM-AA) (UiO-66-P(NIPAM-AA)) and released under regulated conditions utilizing UiO-66-P (NIPAM-AA); (**c**) procaine amide’s sequential release and halt exhibited by UiO-66-P/NIPAM-AA in water are time-dependent and affected by pH. Reproduced with permission from references [146,147].

**Figure 13 pharmaceutics-15-00931-f013:**
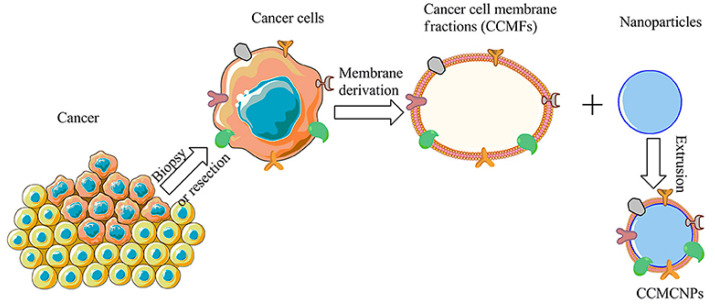
Diagram depicting the processes required for the synthesis of tumor cell membrane-coated nanomaterials. Reproduced with permission from [153]. Copyright 2020 Frontiers in Oncology.

**Figure 14 pharmaceutics-15-00931-f014:**
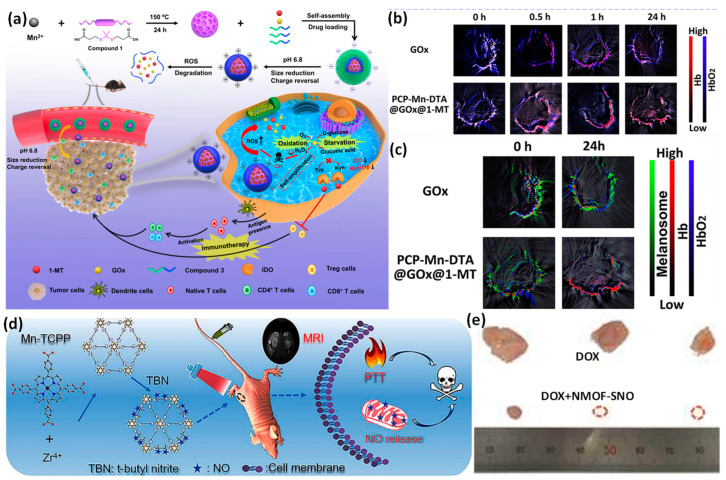
(**a**) Synthetic approach and graphical depiction of PCP-Mn-DTA@GOx@1-MT nanomaterial for combination starvation, immunotherapy, and oxidation; (**b**) hemoglobin (HB) and oxygenated hemoglobin (HBO_2_) photoacoustic recordings; (**c**) melanin signals in mouse tumor sites following intratumoral injecting with PCP-Mn-DTA@GOx@1-MT and GOx for multiple time frames; (**d**) a plan for creating MOF-SNO nanocomposite, releasing nitric oxide when exposed to NIR light, and using photothermal therapy; (**e**) pictures of the tumors in each group following treatment. Reproduced with permission from references [160,163].

**Table 1 pharmaceutics-15-00931-t001:** Physicochemical properties and applications of nanomaterials for drug delivery.

Kind of Material	Size (nm)	Shape	Surface Area (m^2^/g)	Properties	Application	Ref.
Silica	100–108	Nanoparticles	1156.4	The photothermal heating effect, efficient endocytosis, (pH, NIR irradiation)-responsive, anchor effect	Chemo-photothermal therapy, active targeting	[32]
Carbon dots	4	Nanodots	-	Electrostatic interactions, pH-dependent release	Chemotherapy	[33]
Silicon	100	Nanoparticles	1407	(pH and NIR)-responsiveness, mitochondrial targeting	Fluorescent image, chemo-photothermal therapy, active targeting	[34]
Liposome	165	Nanoparticles	-	X-ray-triggered liposomes	Chemotherapy, radiotherapy, photodynamic therapy	[35]
Magnetic-gold	11–29	Nanoparticles	-	Multifunctional magnetic gold, controlled-release manner	Passively magnetic targeting; chemmophotothermal therapy; magnetic resonance imaging (MRI)	[36]
Carbon nanotubes	0.4–2/2–100	Cylindrical roll	232.5	π-π stacking, electrostatic interaction, pharmaco-toxicological properties	Chemotherapy	[37]
Hydrogel	35–60	Sphere NPs	-	Thermo-sensitive micelles, reversible sol–gel transition,	Chemotherapy	[38]
Protein	28	Monodisperse nano-scaffold	-	Receptor-mediated internalization, fluorescent image	Chemotherapy, active targeting	[39]
ZIF-8	50–160	Dodecahedral	1925	π–π stacking, hydrogen bonding, electrostatic interactions, fluorescent imaging, and pH-responsive drug release	Chemotherapy, passive targeting	[40]

**Table 2 pharmaceutics-15-00931-t002:** A summary of documented MOFs for the delivery of medicinal substances [59].

Therapeutic Drug	MOFs	Organic Linker	Metal Ion	Drug Encapsulation Method	Ref.
*1. Anti-inflammatory and analgesics drugs*					
Ibuprofen	MIL-100	1,3,5-benzene tricarboxylic acid (BTC)	Cr^3+^	Post-synthetic (PS) encapsulation	[60]
Ibuprofen	MIL-101	1,4-benzene dicarboxylic acid (BDC)	Cr^3+^	PS encapsulation	[60]
Ibuprofen	MIL-53	BDC	Fe^3+^, Cr^3+^	PS encapsulation	[61]
Curcumin, Sulindac	MOF-5	BDC	Zn^2+^	PS encapsulation	[62]
Diclofenac sodium	ZJU-800	F-H2PDA	Zr^2+^	PS encapsulation	[63]
*2. Antiviral and antibacterial drugs*					
Cidofovir	MIL-101-NH2	2-amino-BDC	Fe^3+^	PS encapsulation	[64]
Nalidixic acid	Bio-MOF	Nalidixic acid	Mg^2+^, Mn^2+^	Direct assembly	[65]
Vancomycin	MIL-53	BDC	Fe^3+^	PS encapsulation	[66]
Ciprofloxacin	UiO-66	BDC	Zr^4+^	PS encapsulation	[67]
Gentamicin	ZIF-8	2-methyl imidazolate	Zn^2+^	PS encapsulation	[68]
Ciprofloxacin	ZIF-8	2-methyl imidazolate	Zn^2+^	PS encapsulation	[69]
Ceftazidime	ZIF-8	2-methyl imidazolate	Zn^2+^	One-pot synthesis (OPS)	[70]
Tetracycline	ZIF-8	2-methyl imidazolate	Zn^2+^	OPS	[71]
Enrofloxacin, Florfenicol	γ-CD-MOF	Cyclodextrin	K^+^	PS encapsulation	[72]
*3. Anti-cancer drugs*					
Nimesulide	HKUST-1	BTC	Cu^2+^	PS encapsulation	[73]
Busulfan	MIL-100	BTC	Fe^3+^	PS encapsulation	[64]
Doxorubicin	MIL-100	BTC	Fe^3+^	PS encapsulation	[74]
Doxorubicin	MIL-89	Muconic acid	Fe^3+^	PS encapsulation	[64]
Oridonin	MOF-5	BDC	Zn^2+^	PS encapsulation	[75]
Cisplatin	NCP-1	Disuccinatocisplatin	Tb^3+^	Direct assembly	[76]
Methotrexate	PCN-221	TCPP	Zr^4+^	PS encapsulation	[77]
Alendronate	UiO-66	BDC	Zr^4+^	Covalent bonding	[78]
Doxorubicin	ZIF-67	2-methyl imidazolate	Co^2+^	OPS	[79]
5-Fluoro uracil	ZIF-67	Imidazole-2-carboxaldehyde	Co^2+^	PS encapsulation	[80]
Doxorubicin	ZIF-67	Imidazole-2-carboxaldehyde	Co^2+^	Covalent bonding	[80]
5-Fluoro uracil	ZIF-8	2-methyl imidazolate	Zn^2+^	PS encapsulation	[81]
Camptothecin	ZIF-8	2-methyl imidazolate	Zn^2+^	OPS	[82]
Doxorubicin	ZIF-8	2-methyl imidazolate	Zn^2+^	OPS	[83]
3-Methyl adenine	ZIF-8	2-methyl imidazolate	Zn^2+^	OPS	[84]
Doxorubicin, Camptothecin, Daunomycin	Zn(bix)	bix	Zn^2+^	OPS	[85]
*4. Peptides, Proteins, and enzymes*					
Insulin	NU-1000	4,4′,4″,4‴-(pyrene-1,3,6,8-tetrayl)tetrabenzoic acid	Zr^4+^	PS encapsulation	[86]
Glucose oxidase	Cu-TCCP(Fe)	TCPP(Fe)	Cu^2+^	Surface attachment	[87]
Insulin	MIL-100	1,3,5-benzene tricarboxylic acid	Fe^3+^	PS encapsulation	[88]
Myoglobin	MOF-74	2,5-dioxido terephthalate	Zn^2+^, Mg^2+^	PS entrapment	[89]
Tyrosinase	PCN-333	TATB	Al^3+^	PS entrapment	[90]
Cytochrome c	Tb-meso MOF	Triazine-1,3,5-tribenzoic acid	Tb^3+^	PS entrapment	[91]
Microperoxidase-11	Tb-meso MOF	Triazine-1,3,5-tribenzoic acid	Tb^3+^	PS entrapment	[92]
Glucose oxidase	ZIF-8	2-methyl imidazolate	Zn^2+^	OPS	[93]
Horseradish peroxidase	ZIF-8	2-methyl imidazolate	Zn^2+^	OPS	[94]
Hemoglobin, Glucose oxidase	ZIF-8	2-methyl imidazolate	Zn^2+^	Biomimetic mineralization	[95]
Melittin	ZIF-8	2-methyl imidazolate	Zn^2+^	OPS	[96]
Catalase	ZIF-90	Imidazole-2-carboxaldehyde	Zn^2+^	OPS	[97]
*5. Antibodies and antigens*					
αCD47	Hf-DBP	5,15-di(p-benzoato) porphyrin	Hf^4+^	Surface attachment	[98]
H-IgG,	ZIF-90	Imidazole-2-carboxaldehyde	Zn^2+^	OPS	[99]
G-IgG	ZIF-90	Imidazole-2-carboxaldehyde	Zn^2+^	OPS	[99]
Nivolumab	ZIF-8	2-methyl imidazolate	Zn^2+^	Biomimetic mineralization	[95]
Ovalbumin	ZIF-8	2-methyl imidazolate	Zn^2+^	One-pot synthesis	[100]
anti-EpCAM	MIL-100	BTC	Fe^3+^	Surface attachment	[101]
Ovalbumin	UiO-AM	BDC, 2-amino-BDC	Zr^4+^	Surface attachment	[102]
Ovalbumin	Al-MOF	BDC, 2-amino-BDC	Al^3+^	OPS	[103]
*6. Nucleotides and Nucleic Acids*					
siRNA	MIL-101	BDC	Fe^3+^	Covalent-linkage	[104]
Terminal phosphate modified oligo-nucleotides	UiO-66	BDC	Zr^4+^	Covalent linkage	[105]
Plasmid DNA	ZIF-8	2-methyl imidazolate	Zn^2+^	OPS	[106]
*7. Carbohydrates*					
Heparin, Hyaluronic acid	MAF-7	2-methyl imidazolate	Zn^2+^	Biomimetic mineralization	[107]
Meglumine, Carboxylate dextran	ZIF-8	2-methyl imidazolate	Zn^2+^	OPS	[108]

**Table 3 pharmaceutics-15-00931-t003:** Advantages and disadvantages of MOFs nanomaterial and example for cancer treatment.

Synthesis Method	MOF	Advantages	Shortcomings	Ref.
Electrochemical	Zr-MOF Zn-MOF	High loading capacity and controlled release of anticancer drugs. Ability to accommodate imaging agents for theranostic applications. Selective targeting and enhanced permeability to tumor sites.	Potential toxicity and biocompatibility issues. Complex synthesis and functionalization procedures. Limited clinical trials and regulatory approvals.	[21,176]
Solvothermal	Zn-MOF-74 MIL-100 (Fe) UiO-66	High purity and crystallinity. Control over size and shape. Ability to incorporate functional groups.	Long reaction time Discontinuity of the process. Inhomogeneity of heating.	[21,177,178]
Ultrasonic	Cu-MOF Fe-MOF	High surface area and porosity. Flexible and tunable chemical structure and architecture. Ability to capture and degrade. Ability to carry anticancer drugs and imaging agents.	Low mechanical and thermal stability. Difficult to recycle and reuse. Potential toxicity to living environments. Possible immune reaction or poor biocompatibility in the human body.	[179,180]
Diffusion	Zr-MOF MIL-100	Large surface area and porosity that can accommodate various drugs and imaging agents. High chemical stability and biocompatibility. Easily functionalized and modified with different ligands and nanoparticles. Enable controlled drug release by external stimuli such as pH, temperature, and light. Enhance the therapeutic efficacy and reduce the side effects of drugs by targeting specific cancer cells.	Low solubility and dispersibility in biological fluids. Induce immune responses or toxicity in some cases. Limited loading capacity or release rate for some drugs. Suffer from aggregation or degradation in vivo.	[11,181]

## Data Availability

Not applicable.

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
