# Peer review of "Utilization of Functionalized Metal–Organic Framework Nanoparticle as Targeted Drug Delivery System for Cancer Therapy"

_pharmaceutics, 2023, doi:10.3390/pharmaceutics15030931_

Round 1

Reviewer 1 Report

This paper titled " Utilization of Functionalized Metal-Organic Framework Nanoparticle as Targeted Drug Delivery System for Cancer Therapy" aims to summarize the recent progress of MOF carriers for tumor therapy, including their principle, synthesis and properties. This review paper is meaningful and timely. However, there are still some problems. The manuscript in its current form requires major revisions as follows:

1. This paper lacks a chart to summarize the logical thinking, structure and key points of the whole paper.

2. The abstract part is a little short, which should be a high summary of the whole content. It can supplement some harm of cancer to human society at the beginning, and then introduce the materials and treatment methods.

3. The advantages and shortcomings of MOFs should be listed in a table.

4. Future development strategy of MOF based materials towards cancer therapy should be also emphasized in the conclusion part.

5. There are some problems in the format of the article, especially the reference format. Please edit it carefully following the requirements of the journal. For example, the volume and page missed in the ref 4. The journal names were abbreviated in ref 12, which was inconsistent with other references. The journal names “Angewandte chemie” in ref 39 should be modified as the international version “Angewandte Chemie International Edition” in ref 41.

6. More related references should be cited: 1. Angew. Chem. Int. Ed., 2022, 61, e202214794. 2. Angew. Chem. Int. Ed., 2023, DOI: 10.1002/anie.202215307. 3. ACS Mater. Lett., 2023, 5, 466-472. 4. J. Am. Chem. Soc., 2023, 145, 1955-1963. 5. Chem. Eng. J., 2022, 435, 135046. 6. J. Mater. Sci. Technol., 2023, 141, 11-20. 6. Dyes and Pigments, 2022, 206, 110655.

Author Response

March 03, 2023                                            

Journal: Pharmaceutics

Manuscript ID: pharmaceutics-2195965

Type of manuscript: Review

Title: Utilization of Functionalized Metal-Organic Framework Nanoparticle as Targeted Drug Delivery System for Cancer Therapy

Dear Editor,

Thank you for allowing us to submit a revised draft of the manuscript “Utilization of Functionalized Metal-Organic Framework Nanoparticle as Targeted Drug Delivery System for Cancer Therapy” for publication in Pharmaceutics. We appreciate the time and effort that you and the reviewers dedicated to providing feedback on our manuscript and are grateful for the insightful comments and valuable improvements to our paper.

We have implemented the majority of the reviewers' recommendations. Those changes are highlighted within the manuscript. Please see below, in blue, for a point-by-point response to the reviewers’ comments and concerns. All page numbers refer to the revised manuscript file with tracked changes.

Best regards,

Dr. Tran Anh Vy

Institute of Applied Technology and Sustainable Development

Nguyen Tat Thanh University, Ho Chi Minh city, Viet Nam

Email: tavy@ntt.edu.vn,  tranhvy@gmail.com

Dr. Vo Ngoc Linh Giang

Faculty of Pharmacy, University of Medicine and Pharmacy at Ho Chi Minh City

Ho Chi Minh City, Viet Nam

Email: vongoclinhgiang@uphcm.edu.vn
Response to Reviewers
and Editor

Reviewer #1

This paper titled " Utilization of Functionalized Metal-Organic Framework Nanoparticle as Targeted Drug Delivery System for Cancer Therapy" aims to summarize the recent progress of MOF carriers for tumor therapy, including their principle, synthesis, and properties. This review paper is meaningful and timely. However, there are still some problems. The manuscript in its current form requires major revisions as follows:

  1. This paper lacks a chart to summarize the logical thinking, structure, and key points of the whole paper.

Author response: Thank you for pointing this out. We agree with the reviewer and have made the change by including a graphical representation to summarize the logical thinking on page 3.

  1. The abstract part is a little short, which should be a high summary of the whole content. It can supplement some harm of cancer to human society at the beginning, and then introduce the materials and treatment methods.

Author response: Thank you for pointing this out. We agree with the reviewer and now include the societal damage caused by cancer before we go into the specifics of the materials and therapies used to combat the disease in the abstract part.

  1. The advantages and shortcomings of MOFs should be listed in a table.

Author response: We have added the advantages and disadvantages of MOFs in table 3.

  1. Future development strategy of MOF based materials towards cancer therapy should be also emphasized in the conclusion part.

Author response: As suggested by the reviewer. We have emphasized the future development strategy of MOF-based materials aimed at cancer therapy in the conclusion part.

  1. There are some problems in the format of the article, especially the reference format. Please edit it carefully following the requirements of the journal. For example, the volume and page missed in the ref 4. The journal names were abbreviated in ref 12, which was inconsistent with other references. The journal names “Angewandte chemie” in ref 39 should be modified as the international version “Angewandte Chemie International Edition” in ref 41.

Author response: We appreciate the reviewer's detailed and careful comments. We have double-checked all references, for format and information and follow the format of Pharmaceutics. We have also corrected these errors.

  1. More related references should be cited: 1. Angew. Chem. Int. Ed., 2022, 61, e202214794. 2. Angew. Chem. Int. Ed., 2023, DOI: 10.1002/anie.202215307. 3. ACS Mater. Lett., 2023, 5, 466-472. 4. J. Am. Chem. Soc., 2023, 145, 1955-1963. 5. Chem. Eng. J., 2022, 435, 135046. 6. J. Mater. Sci. Technol., 2023, 141, 11-20. 6. Dyes and Pigments, 2022, 206, 110655.

Author response:  We have added references such as a recommendation from a reviewer according to the latest update from the manuscript.

Reviewer 2 Report

The review is of considerable interest and well done. I recommend it to be published after a minor revision.

1. The introduction is adequate, however, the role of heterogeneous catalysts doesn’t clear, and the authors should give some examples on these MOF and also, should insert the potential application of MOF.

2. The Authors should also proofread their manuscript (some spelling and grammar errors).

3. The author should better improve the beauty and quality of the figures in the manuscript.

4.Some publications are suggested to refer to improve the quality of the manuscript, such as: International Journal of Nano and Material Sciences, 2018, 7(1): 31-42, https://doi.org/10.1007/s10562-022-04026-y, https://doi.org/10.1016/j.ceramint.2022.10.029.

5. The conclusion is too long and also not targeted to the important aspects described in the review; please rephrase it.

Author Response

March 03, 2023                                            

Journal: Pharmaceutics

Manuscript ID: pharmaceutics-2195965

Type of manuscript: Review

Title: Utilization of Functionalized Metal-Organic Framework Nanoparticle as Targeted Drug Delivery System for Cancer Therapy

Dear Editor,

Thank you for allowing us to submit a revised draft of the manuscript “Utilization of Functionalized Metal-Organic Framework Nanoparticle as Targeted Drug Delivery System for Cancer Therapy” for publication in Pharmaceutics. We appreciate the time and effort that you and the reviewers dedicated to providing feedback on our manuscript and are grateful for the insightful comments and valuable improvements to our paper.

We have implemented the majority of the reviewers' recommendations. Those changes are highlighted within the manuscript. Please see below, in blue, for a point-by-point response to the reviewers’ comments and concerns. All page numbers refer to the revised manuscript file with tracked changes.

Best regards,

Dr. Tran Anh Vy

Institute of Applied Technology and Sustainable Development

Nguyen Tat Thanh University, Ho Chi Minh city, Viet Nam

Email: tavy@ntt.edu.vn,  tranhvy@gmail.com

Dr. Vo Ngoc Linh Giang

Faculty of Pharmacy, University of Medicine and Pharmacy at Ho Chi Minh City

Ho Chi Minh City, Viet Nam

Email: vongoclinhgiang@uphcm.edu.vn

Reviewer #2

The review is of considerable interest and well done. I recommend it to be published after a minor revision.

  1. The introduction is adequate, however, the role of heterogeneous catalysts doesn’t clear, and the authors should give some examples on these MOF and also, should insert the potential application of MOF.

Author response: Dear reviewer, as far as we're aware, MOFs are excellent candidates for heterogeneous catalysts. Unfortunately, the article is structured to provide brief examples of a wide range of MOF functions in cancer target therapy treatment, it is difficult for us to go further into the topic of MOF heterogeneous catalysts. It is our empathetic goal that the heterogeneous catalysts of MOF will emerge as the primary focus of our upcoming review. We would want to express our gratitude to you for your recommendation, we have made an effort to recreate the opening paragraph of the introduction on page 1 so that it is identical to what you have mentioned.

  1. The Authors should also proofread their manuscript (some spelling and grammar errors).

Author response: We've proofread the entire manuscript and correct errors such as spelling and grammar.

  1. The author should better improve the beauty and quality of the figures in the manuscript.

Author response: We double-checked the images used in the manuscript, and made sure they met the journal's requirements.

  1. Some publications are suggested to refer to improve the quality of the manuscript, such as: International Journal of Nano and Material Sciences, 2018, 7(1): 31-42, https://doi.org/10.1007/s10562-022-04026-y, https://doi.org/10.1016/j.ceramint.2022.10.029.

Author response: As suggested by the reviewer. We have added more information from two suggested papers on page 1.

  1. The conclusion is too long and also not targeted to the important aspects described in the review; please rephrase it.

Author response: As suggested by the reviewer. We have emphasized the future development strategy of MOF-based materials aimed at cancer therapy in the conclusion part.

Reviewer 3 Report

Title: “Utilization of Functionalized Metal-Organic Framework Nanoparticle as Targeted Drug Delivery System for Cancer Therapy”

Journal: Pharmaceutics (ISSN 1999-4923)

Manuscript Number: pharmaceutics-2195965

In this review article, Tran and colleagues reviewed the utilization of functionalized metal-organic framework nanoparticles for cancer therapy. Despite the fact that this paper is well written, it is not a novel contribution, and there have been many other papers of a similar nature published in the past. It is therefore recommended that the manuscript be published only after adding some new applications or novelty.

Here are some suggestions for improving the paper

  1. Author need to improve the introduction and compare MOF with other peptide materials.

https://pubs.acs.org/doi/10.1021/acsanm.2c01632

https://www.mdpi.com/1999-4923/15/2/345

  1. There are some recent MOF applications in cancer that need to be included
  2. The mechanism of drug release and loading for targeted cancer cells/tissues, as well as the long-term stability of the MOF loaded drug, needs to be discussed in more detail.
  3. The introduction, the results, and the conclusion should be thoroughly revised.

Author Response

March 03, 2023                                            

Journal: Pharmaceutics

Manuscript ID: pharmaceutics-2195965

Type of manuscript: Review

Title: Utilization of Functionalized Metal-Organic Framework Nanoparticle as Targeted Drug Delivery System for Cancer Therapy

Dear Editor,

Thank you for allowing us to submit a revised draft of the manuscript “Utilization of Functionalized Metal-Organic Framework Nanoparticle as Targeted Drug Delivery System for Cancer Therapy” for publication in Pharmaceutics. We appreciate the time and effort that you and the reviewers dedicated to providing feedback on our manuscript and are grateful for the insightful comments and valuable improvements to our paper.

We have implemented the majority of the reviewers' recommendations. Those changes are highlighted within the manuscript. Please see below, in blue, for a point-by-point response to the reviewers’ comments and concerns. All page numbers refer to the revised manuscript file with tracked changes.

Best regards,

Dr. Tran Anh Vy

Institute of Applied Technology and Sustainable Development

Nguyen Tat Thanh University, Ho Chi Minh city, Viet Nam

Email: tavy@ntt.edu.vn,  tranhvy@gmail.com

Dr. Vo Ngoc Linh Giang

Faculty of Pharmacy, University of Medicine and Pharmacy at Ho Chi Minh City

Ho Chi Minh City, Viet Nam

Email: vongoclinhgiang@uphcm.edu.vn

Reviewer #3

In this review article, Tran and colleagues reviewed the utilization of functionalized metal-organic framework nanoparticles for cancer therapy. Despite the fact that this paper is well written, it is not a novel contribution, and there have been many other papers of a similar nature published in the past. It is therefore recommended that the manuscript be published only after adding some new applications or novelty. Here are some suggestions for improving the paper

  1. Author need to improve the introduction and compare MOF with other peptide materials. https://pubs.acs.org/doi/10.1021/acsanm.2c01632; https://www.mdpi.com/1999-4923/15/2/345.

Author response: As suggested by the reviewer. We have added more information from two suggested papers on page 5.

  1. There are some recent MOF applications in cancer that need to be included.

Author response: Recent applications of MOF and material modification MOF applications in cancer have been added by us in the Introduction section in the latest update.

  1. The mechanism of drug release and loading for targeted cancer cells/tissues, as well as the long-term stability of the MOF loaded drug, needs to be discussed in more detail.

Author response:

Indeed, we have covered things like the mechanism of drug release and loading for targeted cancer cells/tissues, and detailed analysis in each case throughout this study includes factors such as Physicochemical targeting, MOFs-based bionic immune, MOF-based nanotherapeutics, and MOF Multi-Targeted Response. It has been shown in the items below.

4.3. Physicochemical targeting cancer therapy by MOF nanomaterials

4.3.1. Light-Responsive targeted cancer therapy by MOF nanomaterials

4.3.2. pH-Responsive targeted cancer therapy by MOF nanomaterials

4.3.3. Magnetic-Field-Responsive targeted cancer therapy by MOF nanomaterials

4.3.4. Redox responsive and targeted cancer therapy by MOF nanomaterials

4.3.5. Thermosensitive MOFs for targeted cancer therapy

4.4 MOFs-based bionic immune for targeted cancer therapy

4.5. MOF-based nanotherapeutics as gene delivery for targeted cancer therapy

4.6 MOF Multi-Targeted Response for cancer therapy

However, the issue of the long-term stability of the MOF-loaded drug needs further discussion. And we have added this issue in the "5.2. Drug release before reaching the target cancer" section of the last update. We hope that our additional efforts will satisfy the reviewer's request.

  1. The introduction, the results, and the conclusion should be thoroughly revised

Author response: We have thoroughly checked the sections as requested by the reviewer, and have edited, supplemented, and changed following the general content of this review, thereby appealing to the readers. These sections have been shown in the latest update.

Round 2

Reviewer 1 Report

1. There are still some problems in the format of the references. For example, the Ref 155 has lost its page numbers.

2. Why didn't the author cite these releated references? Does this mean that the author does not acknowledge these works?  1. Angew. Chem. Int. Ed., 2022, 61, e202214794. 2. Angew. Chem. Int. Ed., 2023, 135, e202215307. 3. ACS Mater. Lett., 2023, 5, 466-472. 4. J. Am. Chem. Soc., 2023, 145, 1955-1963. 5. Chem. Eng. J., 2022, 435, 135046. 6. J. Mater. Sci. Technol., 2023, 141, 11-20. 6. Dyes and Pigments, 2022, 206, 110655.

Author Response

March 08, 2023                                            

Journal: Pharmaceutics

Manuscript ID: pharmaceutics-2195965

Type of manuscript: Review

Title: Utilization of Functionalized Metal-Organic Framework Nanoparticle as Targeted Drug Delivery System for Cancer Therapy

Dear Editor,

Thank you for allowing us to submit a revised draft of the manuscript “Utilization of Functionalized Metal-Organic Framework Nanoparticle as Targeted Drug Delivery System for Cancer Therapy” for publication in Pharmaceutics. We appreciate the time and effort that you and the reviewers dedicated to providing feedback on our manuscript and are grateful for the insightful comments and valuable improvements to our paper.

We have implemented the majority of the reviewers' recommendations. Those changes are highlighted within the manuscript. Please see below, in blue, for a point-by-point response to the reviewers’ comments and concerns. All page numbers refer to the revised manuscript file with tracked changes.

Best regards,

Dr. Tran Anh Vy

Institute of Applied Technology and Sustainable Development

Nguyen Tat Thanh University, Ho Chi Minh city, Viet Nam

Email: tavy@ntt.edu.vn,  tranhvy@gmail.com

Dr. Vo Ngoc Linh Giang

Faculty of Pharmacy, University of Medicine and Pharmacy at Ho Chi Minh City

Ho Chi Minh City, Viet Nam

Email: vongoclinhgiang@uphcm.edu.vn
Response to Reviewers and Editor

Reviewer #1

  1. There are still some problems in the format of the references. For example, the Ref 155 has lost its page numbers.

Auth We have re-checked and supplemented the missing information in the reference. Thanks for the detailed review by the reviewer.or Response:

  1. Why didn't the author cite these releated references? Does this mean that the author does not acknowledge these works? 1. Angew. Chem. Int. Ed., 2022, 61, e202214794. 2. Angew. Chem. Int. Ed., 2023, 135, e202215307. 3. ACS Mater. Lett., 2023, 5, 466-472. 4. J. Am. Chem. Soc., 2023, 145, 1955-1963. 5. Chem. Eng. J., 2022, 435, 135046. 6. J. Mater. Sci. Technol., 2023, 141, 11-20. 6. Dyes and Pigments, 2022, 206, 110655.

Author Response:

We are so regretful. Because of our misunderstanding, we are unable to adequately respond to your recommendation. We are grateful to you for this additional opportunity. Within the text of the paper, we have made citations of the information that you suggest.

Reviewer #3

Tran and colleagues have improved the manuscript now but some minor suggestion need to improve It is therefore recommended that the manuscript can be published after adding my first concerned.

Here are missing suggestions, which authors did not answer fully.

Author need to improve the introduction and compare MOF with other peptide materials.

https://pubs.acs.org/doi/10.1021/acsanm.2c01632 

https://pubs.rsc.org/en/content/articlelanding/2021/bm/d0bm01839b

https://www.mdpi.com/1999-4923/15/2/345 

https://pubs.acs.org/doi/10.1021/acs.jpcb.2c06751

https://pubs.rsc.org/en/content/articlehtml/2021/cc/d1cc05157a

After this response, editor can be accept this manuscript.

Author Response:

We highly appreciate the reviewers’ insightful and helpful comments on our manuscript. The introduction and compare MOF with other peptide materials is essential. We have made an effort to create a paragraph in the introduction on page 2 so that it is comparable to what you have mentioned.

Reviewer 3 Report

Title: “Utilization of Functionalized Metal-Organic Framework Nanoparticle as Targeted Drug Delivery System for Cancer Therapy”

Journal: Pharmaceutics (ISSN 1999-4923)

Manuscript Number: pharmaceutics-2195965

Tran and colleagues have improved the manuscript now but some minor suggestion need to improve It is therefore recommended that the manuscript can be published after adding my first concerned.

Here are missing suggestions, which authors did not answer fully.

  1. Author need to improve the introduction and compare MOF with other peptide materials.

https://pubs.acs.org/doi/10.1021/acsanm.2c01632  

https://pubs.rsc.org/en/content/articlelanding/2021/bm/d0bm01839b

https://www.mdpi.com/1999-4923/15/2/345  

https://pubs.acs.org/doi/10.1021/acs.jpcb.2c06751

https://pubs.rsc.org/en/content/articlehtml/2021/cc/d1cc05157a

After this response, editor can be accept this manuscript.

Author Response

(The authors gave the same response as above.)
